# Erg6 Acts as a Downstream Effector of the Transcription Factor Flo8 To Regulate Biofilm Formation in *Candida albicans*

Xueyang Jin,[a] Xiaoyi Luan,[a] Fei Xie,[b] (ID) Wenqiang Chang,[a] (ID) Hongxiang Lou[a]

[a]Department of Natural Product Chemistry, Key Laboratory of Chemical Biology (Ministry of Education), School of Pharmaceutical Sciences, Cheeloo College of Medicine, Shandong University, Jinan, Shandong Province, China

[b]Department of Pharmacy, Qilu Hospital of Shandong University, Jinan, Shandong Province, China

Xueyang Jin and Xiaoyi Luan contributed equally to this work. Author order was determined by a joint decision of all authors.

**ABSTRACT** The yeast-to-hyphal morphotype transition and subsequent biofilm formation are important virulence factors of *Candida albicans* and are closely associated with ergosterol biosynthesis. Flo8 is an important transcription factor that determines filamentous growth and biofilm formation in *C. albicans*. However, the relationship between Flo8 and regulation of the ergosterol biosynthesis pathway remains elusive. Here, we analyzed the sterol composition of a *flo8*-deficient *C. albicans* strain by gas chromatography-mass spectrometry and observed the accumulation of the sterol intermediate zymosterol, the substrate of Erg6 (C-24 sterol methyltransferase). Accordingly, the transcription level of *ERG6* was reduced in the *flo8*-deficient strain. Yeast one-hybrid experiments revealed that Flo8 physically interacted with the *ERG6* promoter. Ectopic overexpression of *ERG6* in the *flo8*-deficient strain partially restored biofilm formation and *in vivo* virulence in a *Galleria mellonella* infection model. These findings suggest that Erg6 is a downstream effector of the transcription factor Flo8 that mediates the cross talk between sterol synthesis and virulence factors in *C. albicans*.

**IMPORTANCE** Biofilm formation by *C. albicans* hinders its eradication by immune cells and antifungal drugs. Flo8 is an important morphogenetic transcription factor that regulates the biofilm formation and *in vivo* virulence of *C. albicans*. However, little is known about how Flo8 regulates biofilm formation and fungal pathogenicity. Here, we determined that Flo8 directly binds to the promoter of *ERG6* to positively regulate its transcriptional expression. Consistently, loss of *flo8* results in the accumulation of the substrate of Erg6. Moreover, ectopic overexpression of *ERG6* at least partially restores the biofilm formation and virulence of the *flo8*-deficient strain both *in vitro* and *in vivo*. This work provides a new perspective on the metabolic link between transcription factors and morphotypes in *C. albicans*.

**KEYWORDS** *Candida albicans*, Erg6, biofilm formation, Flo8, transcription factors

*Candida albicans* is a common opportunistic pathogen (1, 2). Colonization of mucosal surfaces or the skin by the yeast form of *C. albicans* usually does not cause a host immune response (3). However, when the normal microbiota balance is disturbed or immunity is impaired, *C. albicans* cells switch from the yeast to the hyphal form, which can invade epithelial tissues and escape phagocytosis by macrophages (4, 5). Therefore, the yeast-to-hyphal morphological transition of *C. albicans* is an important virulence factor, and *C. albicans* mutants that cannot form hyphae are typically avirulent (6, 7). Moreover, hyphae are important structural components of *C. albicans* biofilms that are critical for normal biofilm development and maintenance (8). *C. albicans* biofilms form a unique architecture that can withstand host immune defenses and increase resistance to antifungal drugs (9).

**Ad Hoc Peer Reviewer** (ID) Sadri Znaidi, Institut Pasteur de Tunis

Address correspondence to Wenqiang Chang, changwenqiang@sdu.edu.cn, or Hongxiang Lou, louhongxiang@sdu.edu.cn.

The authors declare no conflict of interest.

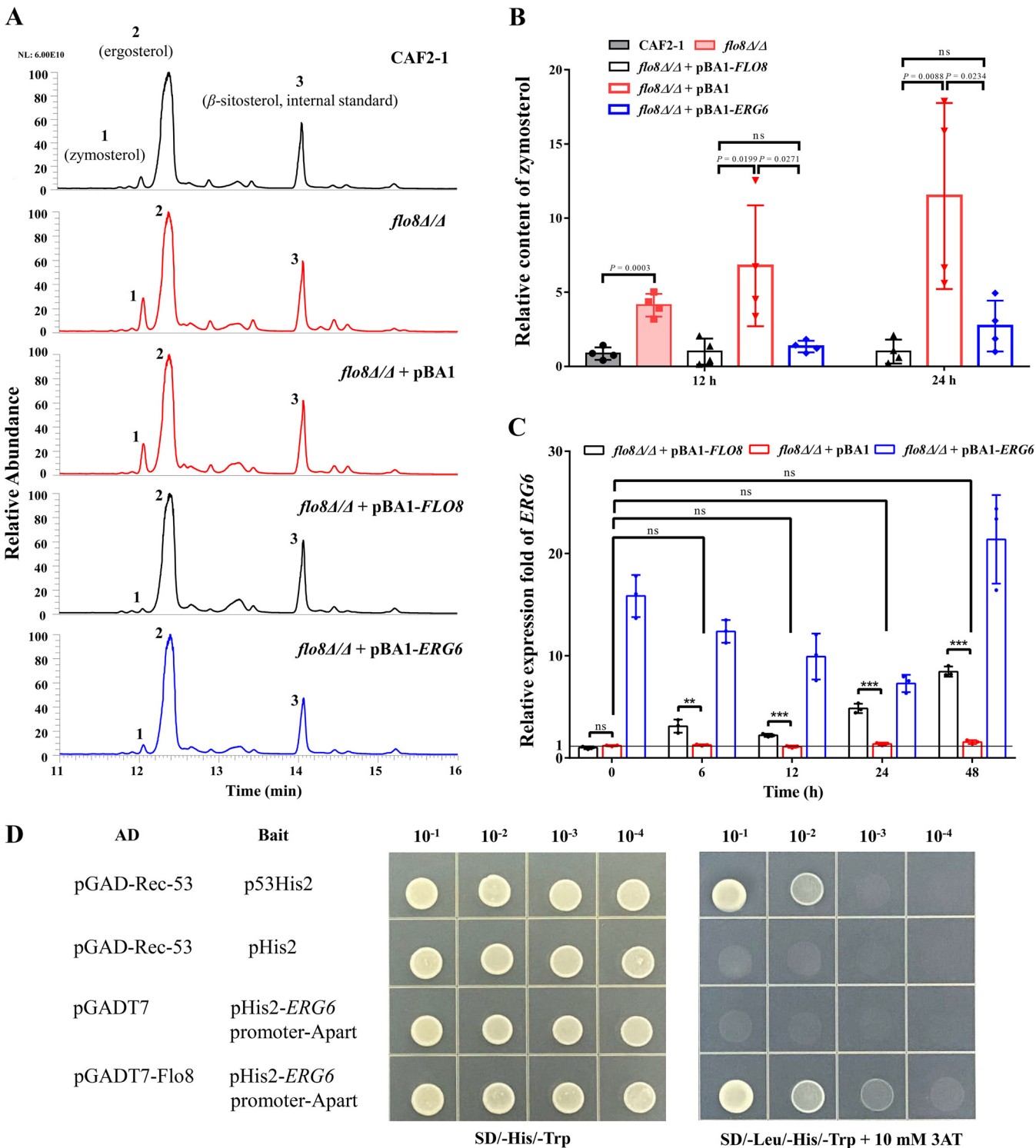

**FIG 1** Erg6 is a downstream protein of the transcription factor Flo8. (A) *C. albicans* CAF2-1, *flo8Δ/Δ*, *flo8Δ/Δ*+pBA1, *flo8Δ/Δ*+pBA1-*FLO8*, or *flo8Δ/Δ*+pBA1-*ERG6* cells were cultured in RPMI 1640 medium for 12 h at 37°C. Sterol components were analyzed by gas chromatography-mass spectrometry. *β*-Sitosterol was added as a standard. Peak 1, zymosterol; peak 2, ergosterol; peak 3, *β*-sitosterol. (B) After calibration using the internal reference, the relative change in zymosterol content in the *flo8Δ/Δ* strain compared with CAF2-1 and *flo8Δ/Δ*+pBA1 strains or the *flo8Δ/Δ*+pBA1-*ERG6* strain compared with the *flo8Δ/Δ*+pBA1-*FLO8* strain was calculated. The bars represent means ± standard deviations. The data obtained from 4 biological replicates are shown. ns, no significance. (C) *C. albicans flo8Δ/Δ*+pBA1, *flo8Δ/Δ*+pBA1-*FLO8*, or *flo8Δ/Δ*+pBA1-*ERG6* cells were incubated in RPMI 1640 medium for 0, 6, 12, 24, or 48 h at 37°C. The transcription levels of *ERG6* in *flo8Δ/Δ*+pBA1-*FLO8*, *flo8Δ/Δ*+pBA1, or *flo8Δ/Δ*+pBA1-*ERG6* cells as determined by PCR are indicated as the fold change relative to the transcription level of *ERG6* in *flo8Δ/Δ*+pBA1-*FLO8* cells at 0 h. The bars represent the means ± standard deviations from three independent experiments. *, $P < 0.05$; **, $P < 0.01$; ***, $P < 0.001$. ns, no significance. (D) Y1H experiment showing the binding of Flo8 to the promoter of *ERG6*. Yeast cells containing pGAD-Rec-53 and p53His2 were the positive-control group. Yeast cells containing pGAD-Rec-53 and pHis2 or pGADT7 and pHis2-*ERG6*promoter-Apart were the two negative-control groups.

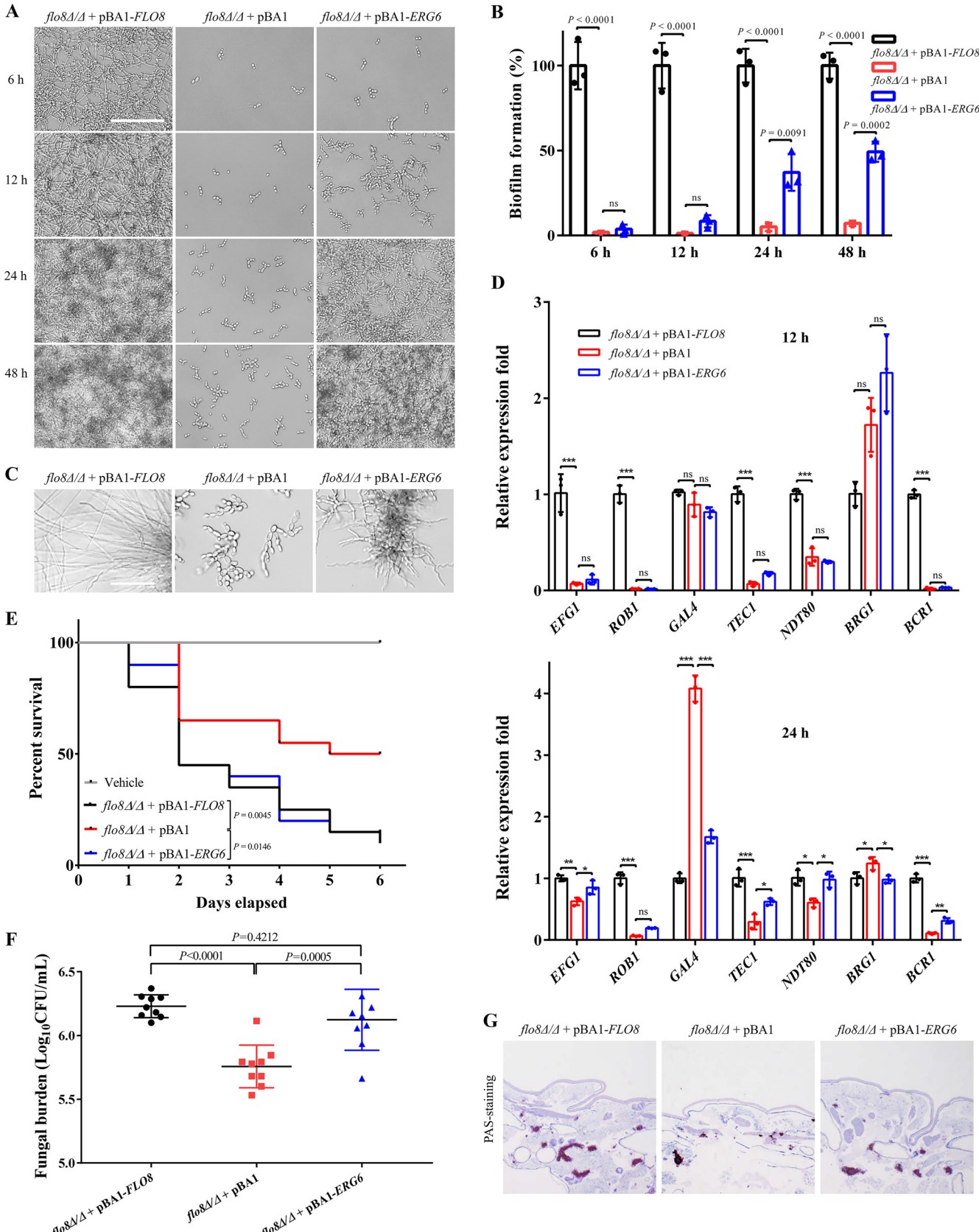

**FIG 2** Overexpression of *ERG6* restores the biofilm formation ability and *in vivo* virulence of the *flo8*-deficient strain. (A and B) *C. albicans flo8Δ/Δ*+pBA1, *flo8Δ/Δ*+pBA1-*FLO8*, or *flo8Δ/Δ*+pBA1-*ERG6* cells (1 × 10⁶) were incubated in RPMI 1640 medium at 37℃. At the indicated times, the cells in the wells were

Sophisticated transcriptional regulatory networks allow *C. albicans* to respond to various host-specific environmental signals, and further exploration of the flexibility of these networks is required to better understand their regulatory mechanisms (10, 11). The transcription factor Flo8 acts downstream of the cAMP-protein kinase A (PKA) pathway and interacts with Efg1 to regulate hyphal growth (12). Flo8 is also required for biofilm development within 48 h (13). *flo8*-deficient strains fail to form filaments and biofilms and are avirulent in systemically infected mouse models (12). Studies of Flo8 have mainly focused on the phenotypic changes in *flo8*-deficient strains or the function of Flo8 (14–16). In contrast, little attention has been given to the genes that Flo8 regulates, particularly those involved in ergosterol biosynthesis.

A previous genome-scale analysis revealed that the ergosterol biosynthesis pathway is crucial for the development of hyphae by *C. albicans* (5). To investigate whether Flo8 regulates hyphal formation via sterol biosynthesis, we first analyzed the sterol composition of the *flo8*Δ/Δ strain and the wild-type strain CAF2-1. The absence of *flo8* resulted in the accumulation of the sterol intermediate zymosterol (Fig. 1A and B), which is converted to fecosterol by C-24 sterol methyltransferase (Erg6). We further compared the sterol compositions of the *flo8*Δ/Δ+pBA1 strain and the reconstituted *flo8*Δ/Δ+pBA1-*FLO8* strain; the accumulation of zymosterol could be observed in the *flo8*-deficient strain after incubation at 37°C for 12 or 24 h (Fig. 1A and B). These results indicated that there is a high probability that the expression of *ERG6* is reduced in the *flo8*-deficient strain. We then performed quantitative PCR (qPCR) to analyze the transcriptional abundance of *ERG6*. Upon induction of hyphal formation in RPMI 1640 medium at 37°C, the expression of *ERG6* in the *flo8*Δ/Δ+pBA1 strain was reduced compared with that in the *flo8*Δ/Δ+pBA1-*FLO8* strain during 48 h of growth (Fig. 1C). A previous study revealed that transcriptional repression or deletion of *ERG6* leads to defects in *C. albicans* hyphal formation (5, 17). Thus, we speculated that Flo8 regulates hyphal or biofilm formation by binding to the putative promoter region of *ERG6*. To test this hypothesis, we conducted a yeast one-hybrid (Y1H) assay in *Saccharomyces cerevisiae* Y187 cells. The recombinant pHIS2-*ERG6*promoter-Apart (−426 bp to −942 bp) or pHIS2-*ERG6*promoter-Bpart (−1 bp to −427 bp) was linearized and used to transform *S. cerevisiae* Y187 for the self-activation test. The results suggested that *S. cerevisiae* Y187 transformed with pHIS2-*ERG6*promoter-Bpart showed strong self-activating activity while *S. cerevisiae* Y187 transformed with pHIS2-*ERG6*promoter-Apart did not show self-activating activity (see Fig. S1 in the supplemental material). As shown in Fig. 1D, yeast cells containing the bait vector carrying pHIS2-*ERG6*promoter-Apart grew on SD/−Leu−Trp−His medium supplemented with 10 mM 3-amino-1,2,4-triazole (3-AT) when cotransformed with the pGADT7-Flo8 construct. In contrast, yeast cells containing the bait vector carrying the promoter region of *ERG6* and the empty pGADT7 vector failed to grow on the same selective medium. These results suggested that Flo8 binds to the promoter of *ERG6 in vivo* and activates expression of a downstream reporter gene in yeast.

To explore whether Flo8 regulates *C. albicans* hyphal development and subsequent biofilm formation by modulating *ERG6* expression, we ectopically overexpressed *ERG6* in a *flo8*Δ/Δ strain. The results of qPCR showed that the transcription of *ERG6* was upregulated in the *flo8*Δ/Δ+pBA1-*ERG6* strain compared with both the *flo8*Δ/Δ+pBA1 strain and the *flo8*Δ/Δ+pBA1-*FLO8* strain during 48 h of growth under hypha-inducing conditions (Fig. 1C).

**FIG 2** Legend (Continued)
washed three times with phosphate-buffered saline and observed under a microscope (bars, 100 $\mu$m). Meanwhile, biofilm formation was quantitatively assessed using the 2,3-bis-(2-methoxy-4-nitro-5-sulfophenyl)-2H-tetrazolium-5-carboxanilide salt reduction assay. The bars represent means $\pm$ standard deviations. ns, no significance. (C) *C. albicans flo8*Δ/Δ+pBA1, *flo8*Δ/Δ+pBA1-*FLO8*, or *flo8*Δ/Δ+pBA1-*ERG6* cells ($1 \times 10^4$) were incubated in RPMI 1640 medium at 37°C. Images were obtained under a microscope after 24 h (bars, 25 $\mu$m). (D) *C. albicans flo8*Δ/Δ+pBA1, *flo8*Δ/Δ+pBA1-*FLO8*, or *flo8*Δ/Δ+pBA1-*ERG6* cells ($1 \times 10^6$) were incubated in RPMI 1640 medium for 12 or 24 h at 37°C. The transcription levels of genes associated with biofilm formation in the *flo8*Δ/Δ+pBA1 or *flo8*Δ/Δ+pBA1-*ERG6* strain as determined by PCR are indicated as the fold change relative to the *flo8*Δ/Δ+pBA1-*FLO8* strain. The bars represent the means $\pm$ standard deviations from three independent experiments. *, $P < 0.05$; **, $P < 0.01$; ***, $P < 0.001$. ns, no significance. (E) *G. mellonella* larvae in each group were infected with $2 \times 10^5$ cells of the specified *C. albicans* strain ($n = 20$ per group). The larvae in the vehicle group were not infected but were injected with an equal volume of phosphate-buffered saline. The number of surviving larvae was recorded each day. The Mantel-Cox test was used to compare differences between groups. (F) Fungal burden of *G. mellonella* infected by *C. albicans* ($n = 9$ per group). (G) The larvae in different groups were fixed in paraformaldehyde and processed for PAS staining. The histopathology of the infected *G. mellonella* larvae was examined by microscopic inspection.

In addition, overexpression of *ERG6* in the *flo8*-deficient strain decreased the level of zymosterol (Fig. 1A and B). We then measured the hyphal development and biofilm formation of *flo8Δ/Δ*+pBA1, *flo8Δ/Δ*+pBA1-*FLO8*, and *flo8Δ/Δ*+pBA1-*ERG6* strains. Ectopic overexpression of *ERG6* did not obviously restore the hyphal elongation of the *flo8Δ/Δ*+pBA1 strain during the first 12 h of growth. However, the defect in biofilm formation of the *flo8Δ/Δ*+pBA1 strain was greatly reversed upon ectopic overexpression of *ERG6* (Fig. 2A and B). The *flo8Δ/Δ*+pBA1-*ERG6* strain also formed a considerable quantity of hyphae at 24 h compared with the *flo8Δ/Δ*+pBA1 strain (Fig. 2C). Consistent with these effects, overexpression of *ERG6* significantly increased the expression of transcription factors that positively regulate filamentation and biofilm formation, including *EFG1*, *TEC1*, *NDT80*, and *BCR1*, after 24 h but not 12 h of induction (Fig. 2D). We further tested the biofilm-forming ability of an *ERG6* knockout strain. The results shown in Fig. S2 suggested the *ERG6* knockout strain was defective in biofilm formation and had changes in the transcript levels of biofilm-regulated genes such as *ROB1*, *GAL4*, *TEC1*, and *BRG1* similar to those in the *FLO8* knockout strain. These results suggested that Erg6 acts as an important downstream effector of Flo8 to regulate biofilm formation. *ERG6* deletion also leads to zymosterol accumulation (18). We deduce that the accumulation of specific sterol intermediates such as zymosterol causes changes in the integrity of the membranes and affects the biofilm formation, but this hypothesis needs further investigation.

Hyphal growth and biofilm formation are important virulence factors in *C. albicans*. To further evaluate the role of *ERG6* in the *in vivo* virulence of *C. albicans*, *Galleria mellonella* larvae were used as an infection model. Infection of larvae with the *flo8Δ/Δ*+pBA1-*ERG6* or *flo8Δ/Δ*+pBA1-*FLO8* strain resulted in a mortality rate exceeding 80% after 6 days (Fig. 2E). In contrast, 50% of the larvae infected with the *flo8Δ/Δ*+pBA1 strain survived more than 6 days (Fig. 2E). Moreover, the fungal burden caused by the *flo8Δ/Δ*+pBA1-*ERG6* strain was similar to that caused by the *flo8Δ/Δ*+pBA1-*FLO8* strain and higher than that caused by the *flo8Δ/Δ*+pBA1 strain (Fig. 2F). Consistent with these observations, histological analysis using periodic acid-Schiff (PAS) staining revealed a higher number of melanized nodules in larvae infected with the *flo8Δ/Δ*+pBA1-*ERG6* strain than in larvae infected with the *flo8Δ/Δ*+pBA1 strain (Fig. 2G). These results suggested that Erg6 is an important effector of Flo8 in the regulation of *C. albicans* virulence.

In summary, we determined that Flo8 directly binds to the putative promoter of *ERG6* to regulate sterol synthesis. Disruption of *flo8* leads to the accumulation of the intermediate zymosterol due to the low expression of *ERG6*, as well as reduced biofilm formation and *in vivo* virulence. Erg6 appears to be an important downstream effector of Flo8 in the regulation of *C. albicans* biofilm formation and *in vivo* virulence. This study provides a perspective for understanding the physiological effects of transcriptional regulators.

## SUPPLEMENTAL MATERIAL

Supplemental material is available online only.
**SUPPLEMENTAL FILE 1**, PDF file, 0.5 MB.

## ACKNOWLEDGMENTS

We thank Jiangye Chen at the Institute of Biochemistry and Cell Biology, University of the Chinese Academy of Sciences, for providing the *C. albicans FLO8*-related strains.

This work was supported by the National Natural Science Foundations (no. 82273975 and 82173703), the Fund for Innovative Team of Shandong University to H.L., the Natural Science Fund for Excellent Young Scholars of Shandong Province of China (ZR2020YQ63), and the Qilu Young Scholars Program of Shandong University to W.C.

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
