## [Reviewer comments · Microbiology Spectrum]

Microbiology Spectrum

Erg6 acts as a downstream effector of the transcription factor Flo8 to regulate biofilm formation in *Candida albicans*

Xueyang Jin, Xiaoyi Luan, Fei Xie, Wenqiang Chang, and Hongxiang Lou

Corresponding Author(s): Wenqiang Chang and Hongxiang Lou, Shandong University Cheeloo College of Medicine

Review Timeline:

Submission Date:	January 25, 2023
Editorial Decision:	March 8, 2023
Revision Received:	April 4, 2023
Accepted:	April 6, 2023

Editor: Renato Kovacs

Reviewer(s): Disclosure of reviewer identity is with reference to reviewer comments included in decision letter(s). The following individuals involved in review of your submission have agreed to reveal their identity: Sadri ZNAIDI (Reviewer #1)

Transaction Report:

DOI: <https://doi.org/10.1128/spectrum.00393-23>

March 8, 2023

Prof. Hongxiang Lou
Shandong University Cheeloo College of Medicine
No. 44 West Wenhua Road
Jinan, Shandong 250012
China

Re: Spectrum00393-23 (Erg6 acts as a downstream effector of the transcription factor Flo8 to regulate biofilm formation in *Candida albicans*)

Dear Prof. Hongxiang Lou:

Link Not Available

Sincerely,

Renato Kovacs

Journals Department
Reviewer comments:

Reviewer #1 (Comments for the Author):

This short report describes the genetic link between *C. albicans* FLO8 and ERG6. Authors show that ERG6 acts downstream of FLO8 (i.e. effector of Flo8). I raise many issues here:

- 1-The report lacks details on how exactly the pBA1 constructs were generated.
- 2-Authors should include data from the parental WT strain, as an additional control for zymosterol levels (the flo8-/- mutant complemented with ADH1p-FLO8 is not sufficient here, because it is still a strain that overexpresses FLO8).
- 3-It is unclear how the biofilm assay was performed. Authors appear to have focused on the initial step of biofilm formation (adhesion step), rather than the mature biofilm steps.

4- The one hybrid assay does not fully demonstrate that Flo8 directly binds to the promoter of ERG6 as stated in line 130 (for example, some interfering molecules in yeast could mediate the interaction) - therefore a complementary approach is needed such as electrophoretic mobility shift assay (EMSA).

5- Line 119: Biofilm formation is a virulence trait that is independent of tissue infection. Here the rationale is not clear. Does *C. albicans* form biofilms on *Galleria*'s tissues?

6- The alteration of ERG6 expression could have some pleiotropic effects, as it is a major player in the integrity of the membranes in yeast (fluidity, permeability, etc.). Authors should provide arguments in favor of specific effects on biofilm growth/hyphal formation rather than a consequence of more general impact mediated, for example, by the integrity of the membrane (i.e. indirect).

Reviewer #2 (Comments for the Author):

In this study, the authors identify Flo8 as a likely transcriptional regulator of Erg6 in *Candida albicans*, and determine a likely role for this interaction in biofilm formation, ergosterol biosynthesis, and a *Galleria* model of virulence. The experiments were well-designed and controlled, and the authors are to be commended on a concise and well-constructed manuscript.

A few minor points the authors might consider addressing:

Line 84: Consider changing "transcriptional expression" to something like "transcript abundance"

The sentence beginning on 87 makes a statement about transcripts of other ergosterol biosynthetic genes not differing in the flo8-/- strain vs its complement, and references figure 1C, but figure 1C only addresses ERG6, not other genes in ergosterol biosynthesis.

Line 104: Consider changing "activates its expression in yeast" to something like "activates expression of a downstream gene in yeast"

The text referring to Fig. 2A in part addresses hyphal growth, but it is difficult to discern the morphology in the images currently in the figure. Perhaps insets of a zoomed in or expanded part of the field could give the reader a better sense of the morphology, particularly at the later timepoints.

There seems to be a small formatting error in the p-values in Fig. 2E.

Staff Comments:

Preparing Revision Guidelines

Please return the manuscript within 60 days; if you cannot complete the modification within this time period, please contact me. If you do not wish to modify the manuscript and prefer to submit it to another journal, please notify me of your decision immediately so that the manuscript may be formally withdrawn from consideration by Microbiology Spectrum.

Reviewer comments:

Reviewer #1 (Comments for the Author):

This short report describes the genetic link between *C. albicans* FLO8 and ERG6. Authors show that ERG6 acts downstream of FLO8 (i.e. effector of Flo8). I raise many issues here:

1- The report lacks details on how exactly the pBA1 constructs were generated.

Response: Thanks for your reminding. We added relevant details on how exactly the pBA1 constructs were generated in the section of Strain construction in the Supplemental Material.

2- Authors should include data from the parental WT strain, as an additional control for zymosterol levels (the flo8^{-/-} mutant complemented with ADH1p-FLO8 is not sufficient here, because it is still a strain that overexpresses FLO8).

Response: We used the wild-type strain CAF2-1 as an additional control to determine the zymosterol level. The new results were showed in Fig. 1A and B and supported that *FLO8* deletion leads to the accumulation of zymosterol, consistent with our previous conclusion.

3- It is unclear how the biofilm assay was performed. Authors appear to have focused on the initial step of biofilm formation (adhesion step), rather than the mature biofilm steps.

Response: Thanks for your reminding. More details about the biofilm assay were provided in the section of Biofilm formation in the Supplemental Material. *C. albicans* is generally considered to form mature biofilms after 24 hours of culture (Refs. 1-3). We observed that the amount of biofilms formed by *flo8Δ/Δ*+pBA1-*ERG6* was significantly different from that of *flo8Δ/Δ*+pBA1 after 24 h of culture. Then we chose two time points including 12 h and 24 h to determine the transcriptional levels of genes related to biofilm formation.

Ref.1. Araújo D, Henriques M, Silva S. Portrait of Candida Species Biofilm Regulatory Network Genes. Trends Microbiol. 2017 Jan;25(1):62-75. doi: 10.1016/j.tim.2016.09.004. Epub 2016 Oct 4. PMID: 27717660.

Ref.2. Fox EP, Bui CK, Nett JE, Hartooni N, Mui MC, Andes DR, Nobile CJ, Johnson AD. An expanded regulatory network temporally controls Candida albicans biofilm formation. Mol Microbiol. 2015 Jun;96(6):1226-39. doi: 10.1111/mmi.13002. Epub 2015 Apr 23. PMID: 25784162; PMCID: PMC4464956.

Ref.3. Mathé L, Van Dijck P. Recent insights into *Candida albicans* biofilm resistance mechanisms. Curr Genet. 2013 Nov;59(4):251-64. doi: 10.1007/s00294-013-0400-3. Epub 2013 Aug 25. PMID: 23974350; PMCID: PMC3824241.

4- The one hybrid assay does not fully demonstrate that Flo8 directly binds to the promoter of ERG6 as stated in line 130 (for example, some interfering molecules in yeast could mediate the interaction) - therefore a complementary approach is needed such as electrophoretic mobility shift assay (EMSA).

Response: We have tried to perform electrophoretic mobility shift assay. However, we failed to obtain the recombinant Flo8 protein using two protein expression vectors including pET-32a and pET-28a-sumo. To exclude the interaction between *ERG6* promoter-Apart and Flo8 that is

potentially interfered by some molecules under experimental conditions, we used yeast cells containing pGADT7 and pHis2-*ERG6*promoter-Apart as a control group. The results showed that *S. cerevisiae* Y187 transformed with pGADT7 (empty vector) and pHis2-*ERG6*promoter-Apart could not grow under SD/-Leu/-His/-Trp + 10 mM 3AT medium but *S. cerevisiae* Y187 transformed with pGADT7-Flo8 and pHis2-*ERG6*promoter-Apart could grow under this condition (Fig. 1D), further supporting the conclusion that Flo8 directly binds to the promoter of *ERG6*.

5-Line 119: Biofilm formation is a virulence trait that is independent of tissue infection. Here the rationale is not clear. Does *C. albicans* form biofilms on *Galleria*'s tissues?

Response: *Galleria mellonella* is a commonly animal model to evaluate the in vivo virulence of *C. albicans* although it is not necessarily to form biofilms on *Galleria*'s tissues. As known, the ability of *C. albicans* in forming biofilms affects its virulence. In addition, we also found *flo8Δ/Δ*+pBA1-*ERG6* exhibited stronger ability of hyphal-formation compared with *flo8Δ/Δ*+pBA1, and hyphal formation is a virulence trait related to tissue infection. In this study, we intended to evaluate the effect of Erg6 on the transcriptional factor Flo8 in terms of in vivo virulence using this wax model.

6-The alteration of *ERG6* expression could have some pleiotropic effects, as it is a major player in the integrity of the membranes in yeast (fluidity, permeability, etc.). Authors should provide arguments in favor of specific effects on biofilm growth/hyphal formation rather than a consequence of more general impact mediated, for example, by the integrity of the membrane (i.e. indirect).

Response: Previous studies have revealed that *ERG6* deletion would lead to defective hyphal formation (Refs. 1 and 2). In the revised version, we tested the biofilm-forming ability of the *erg6* knockout strain. The results showed that lack of Erg6 resulted in defective biofilm formation and alteration of transcription levels of genes closely related to biofilm formation. In our forthcoming published article, we found that deletion of *ERG6* reduced the translational level of the positive transcriptional factor Ume6 and stabilization of the negative transcriptional factor Nrg1, which is closely associated with *C. albicans* biofilm formation.

Moreover, *ERG6* deletion leads to zymosterol accumulation (Ref. 3), which may cause changes in the integrity of the membranes, but the relationship between accumulation of zymosterol and biofilm formation needs thoroughly investigation in the future. We added these results and relevant discussions in the revised manuscript.

Ref.1. O'Meara TR, Veri AO, Ketela T, Jiang B, Roemer T, Cowen LE. Global analysis of fungal morphology exposes mechanisms of host cell escape. *Nat Commun.* 2015 Mar 31;6:6741. doi: 10.1038/ncomms7741. PMID: 25824284; PMCID: PMC4382923.

Ref.2. Dorsaz S, Snäkä T, Favre-Godal Q, Maudens P, Boulens N, Furrer P, Ebrahimi SN, Hamburger M, Allémann E, Gindro K, Queiroz EF, Riezman H, Wolfender JL, Sanglard D. Identification and Mode of Action of a Plant Natural Product Targeting Human Fungal Pathogens. *Antimicrob Agents Chemother.* 2017 Aug 24;61(9):e00829-17. doi: 10.1128/AAC.00829-17. PMID: 28674054; PMCID: PMC5571344.

Ref.3. Jensen-Pergakes KL, Kennedy MA, Lees ND, Barbuch R, Koegel C, Bard M. Sequencing, disruption, and characterization of the *Candida albicans* sterol methyltransferase (*ERG6*) gene:

drug susceptibility studies in *erg6* mutants. *Antimicrob Agents Chemother.* 1998 May;42(5):1160-7. doi: 10.1128/AAC.42.5.1160. PMID: 9593144; PMCID: PMC105764.

Reviewer #2 (Comments for the Author):

In this study, the authors identify Flo8 as a likely transcriptional regulator of Erg6 in *Candida albicans*, and determine a likely role for this interaction in biofilm formation, ergosterol biosynthesis, and a Galleria model of virulence. The experiments were well-designed and controlled, and the authors are to be commended on a concise and well-constructed manuscript.

A few minor points the authors might consider addressing:

Line 84: Consider changing "transcriptional expression" to something like "transcript abundance"

Response: Thanks for your suggestion. We changed it according to your suggestion.

The sentence beginning on 87 makes a statement about transcripts of other ergosterol biosynthetic genes not differing in the *flo8*^{-/-} strain vs its complement, and references figure 1C, but figure 1C only addresses ERG6, not other genes in ergosterol biosynthesis.

Response: Thanks for your reminding. We corrected this error.

Line 104: Consider changing "activates its expression in yeast" to something like "activates expression of a downstream gene in yeast"

Response: Thanks for your suggestion. We changed it according to your suggestion.

The text referring to Fig. 2A in part addresses hyphal growth, but it is difficult to discern the morphology in the images currently in the figure. Perhaps insets of a zoomed in or expanded part of the field could give the reader a better sense of the morphology, particularly at the later timepoints.

Response: We tested the hyphal-forming ability of *flo8Δ/Δ*+pBA1, *flo8Δ/Δ*+pBA1-*ERG6* and *flo8Δ/Δ*+pBA1-*FLO8*, and enlarged images were shown in Figure 2C.

There seems to be a small formatting error in the p-values in Fig. 2E.

Response: Thanks for your reminding. We corrected this error.

April 6, 2023

Prof. Hongxiang Lou
Shandong University Cheeloo College of Medicine
No. 44 West Wenhua Road
Jinan, Shandong 250012
China

Re: Spectrum00393-23R1 (Erg6 acts as a downstream effector of the transcription factor Flo8 to regulate biofilm formation in *Candida albicans*)

Dear Prof. Hongxiang Lou:

Your manuscript has been accepted, and I am forwarding it to the ASM Journals Department for publication. You will be notified when your proofs are ready to be viewed.

Sincerely,

Renato Kovacs
Editor, Microbiology Spectrum
